# The Role of Acceptance and Commitment Therapy in Cardiovascular and Diabetes Healthcare: A Scoping Review

**DOI:** 10.3390/ijerph18158126

**Published:** 2021-07-31

**Authors:** Amineh Rashidi, Lisa Whitehead, Lisa Newson, Felicity Astin, Paramjit Gill, Deirdre A. Lane, Gregory Y. H. Lip, Lis Neubeck, Chantal F. Ski, David R. Thompson, Helen Walthall, Ian D. Jones

**Affiliations:** 1School of Nursing and Midwifery, Joondalup Campus, Edith Cowan University, Perth 6207, Australia; l.whitehead@ecu.edu.au; 2School of Psychology, Faculty of Health, Liverpool John Moores University, Liverpool L3 3AF, UK; l.m.newson@ljmu.ac.uk; 3Centre for Applied Research in Health, University of Huddersfield and Calderdale and Huddersfield National Health Service Foundation Trust, Huddersfield HD1 3DH, UK; F.Astin@hud.ac.uk or; 4Division of Health Sciences, University of Warwick, Coventry CV4 7AL, UK; p.gill.1@warwick.ac.uk; 5Liverpool Centre for Cardiovascular Science, University of Liverpool and Liverpool Heart & Chest Hospital, Liverpool L7 8TX, UK; deirdre.lane@liverpool.ac.uk (D.A.L.); Gregory.Lip@liverpool.ac.uk (G.Y.H.L.); 6School of Health and Social Care, Edinburgh Napier University, Edinburgh EH11 4BN, UK; l.neubeck@napier.ac.uk; 7Integrated Care Academy, University of Suffolk, Ipswich IP4 1QJ, UK; c.ski@uos.ac.uk; 8School of Nursing and Midwifery, Queen’s University Belfast, Belfast BT9 7BL, UK; David.Thompson@qub.ac.uk; 9NIHR Oxford Biomedical Research Centre, The Joint Research Office, Second Floor, OUH Cowley, Unipart House Business Centre, Garsington Road, Oxford OX4 2PG, UK; helen.walthall@ouh.nhs.uk; 10School of Nursing and Allied Health, Faculty of Health, Liverpool John Moores University, Liverpool L2 2ER, UK; I.D.Jones@ljmu.ac.uk

**Keywords:** acceptance and commitment therapy, cardiovascular, diabetes

## Abstract

Acceptance and commitment therapy (ACT) is an adapted form of cognitive behavioural therapy. ACT focuses on how thinking affects behaviour and promotes psychological flexibility. The prevalence of psychological distress among people living with cardiovascular disease (CVD) and/or type 2 diabetes mellitus (T2DM) is high, and ACT may offer an alternative treatment approach. This scoping review explored the use of ACT as an intervention to support adults living with CVD and/or T2DM. A systematic search of the literature resulted in the inclusion of 15 studies. Studies were reviewed using the Joanna Briggs Institute approach to conducting scoping reviews. Most studies (*n* = 13) related to people living with T2DM, and most (*n* = 10) used a pre-post design, four studies were randomised controlled trials, and one was a qualitative study. Eight studies reported an improvement in the outcome(s) assessed post-intervention, suggesting that ACT was an acceptable and valid intervention to support people living with CVD or T2DM. However, studies were underpowered and only limited studies involved people living with CVD. ACT was assessed as a valuable approach to improve a range of patient-reported outcomes for those living with CVD or T2DM, and further research involving robust study designs and larger cohorts are warranted.

## 1. Introduction

The number of people living with cardiovascular disease (CVD), which includes coronary heart disease and stroke, and Type 2 Diabetes Mellitus (T2DM), continues to grow worldwide [1]. More people die from CVD per year than any other condition, accounting for 31% of all deaths in 2016 [2]. Diabetes is also a global issue, with approximately 11% of the adult population known to be living with the condition [3], and the two conditions commonly co-occur [4]. The prevention and management of both conditions are underpinned by the need to adopt a healthy lifestyle through self-management practices relating to diet and exercise, managing stress and medication management [2,5]. Diabetes is a risk factor for cardiovascular disease and there is a high prevalence of co-morbidity [2].

Structured Cardiac Rehabilitation (C.R.) programmes, and diabetes care, focus on improving lifestyle behaviours to reduce smoking, engage with dietary adjustments and increase physical activity. Evidence on the uptake of self-management behaviours to prevent illness and improve health highlight the challenges and variable effectiveness of self-management interventions of long-term conditions [6]. Despite improvements to healthcare services, gaps remain between healthcare advice, patient implementation of advice, management of their condition, and clinical outcomes. Non-attendance at C.R. is associated with suboptimal clinical outcomes; this is applicable for those who never attend and those who drop out. A recent systematic review and interpretative synthesis highlights the complex nature of self-management of T2DM. Making lifestyle changes is challenging, and understanding and accounting for the context in which people live, their social influences, cultural and societal norms, the physical environment, and physiological and psychological factors are important [7].

An interplay exists between patient engagement and implementation of healthcare advice and the individual’s processes in making (un/conscious) decisions, and attending, engaging, and implementing self-management behaviours. In this context, patients are managing behavioural, cognitive, emotional challenges and, potentially, any pre-existing experiences and beliefs as part of their day-to-day management. It is the role of healthcare services to provide interventions to support patients to navigate and manage such psychological challenges. Moreover, an individual’s psychological state may further impact engagement with healthcare advice and self-management behaviours. For people with either CVD and T2DM, depression and anxiety are prevalent comorbidities. These patients may experience additional psychological distress whilst managing and implementing behavioural advice associated with increased mortality and morbidity [8]. In diabetes, this is referred to specifically as “diabetes distress” [9], common in patients with T2DM [10]. Diabetes distress can be influenced by numerous psychological variables, including (though not limited to) illness perceptions and beliefs, self-efficacy, coping strategies, emotional regulation and psychological flexibility [11].

Acceptance and commitment therapy (ACT) is an adapted form of cognitive behavioural therapy (CBT). Unlike Cognitive Behavioural Therapy (CBT) and traditional behavioural-based approaches, ACT focuses on how thinking affects behaviour, and promotes psychological flexibility [12]

In comparison to CBT, ACT does not attempt to change beliefs (e.g., replace negative thoughts with adaptive thoughts), and it does not seek to remove the psychological distress (though this may be a positive outcome). ACT promotes connection to the present moment and engagement in values-based action to help people create a sense of meaning and increase their psychological flexibility [13]

ACT has been used with people living with a wide range of conditions [14], as an intervention to improve self-management [15], to improve medication management behaviours among adolescents living with diabetes [16], and for people living with cardiac disease [17]. Given the prevalence of psychological distress in patients living with CVD and/or T2DM, ACT may be an ideal intervention to integrate into the healthcare offer. Previous reviews have evaluated the application of ACT across (non-specific) long-term conditions [14].

Healthcare services need to be innovative and respond to personalised needs and there is a clear need for improvement in CVD and diabetes services. This review sought to synthesise the research literature on the use of ACT as an intervention to support the management of both conditions.

## 2. Materials and Methods

This review was conducted in accordance with the Joanna Briggs Institute (JBI; May 2020) guidelines for scoping reviews involving narrative synthesis. Scoping reviews are rigorous processes carried out to understand a variety of studies with several methodologies and outcome measures [18].

### 2.1. Search Strategy and Inclusion Criteria

Eight electronic bibliographic databases (Embase, Medline, PsycINFO, CINAHL, Web of Science, Cochrane Library, Scopus, JBI and Google Scholar) were searched in May 2020 (15 May) with an updated search run in November 2020 (20 November). Keywords and Medical Subject Headings (MeSH) terms were used to identify relevant literature. Key terms relevant to ACT and diabetes and cardiovascular care were used (see Appendix A).

### 2.2. Study Selection and Inclusion Criteria

The included studies met the following eligibility criteria: (1) participants, identified as adults diagnosed with type 2 diabetes and/or cardiovascular disease (defined as a group of disorders of the heart and blood vessels); (2) intervention, ACT (3); Design, all study designs; and (4) context, all settings. We excluded studies where full text was unavailable. No date restrictions were implemented for the search period or article publication date. We considered all study designs that examined attributes of interventions, implementation, feasibility, acceptability and the risks or benefits of ACT. Empirical research published in a peer-reviewed journal and any language were included. At the initial stage, all studies were exported to EndNote and duplicates were eliminated. Titles and abstracts were screened against the inclusion criteria by three independent reviewers (L.W., A.R., I.J.). One study was excluded because the full-text study was not available [19].

The abstracts in English of six studies published in Farsi [20,21,22,23,24,25] and two studies published in German [26,27] were screened for inclusion, and all included at this stage. The Google Translator tool was used to translate the Farsi studies [20,21,22,23,24,25] for full-text screening. The accuracy of Google Translate was found to be around 90% [28], and a native speaker (AR) sought to sense-check the accuracy of the verbatim translation and made any corrections to the text as required. The German studies [26,27] were translated into English by a bilingual speaker known to an author and fluent in both languages. Six studies [20,21,22,23,24,25] were translated from Farsi to English by a native Farsi speaker also fluent in English (AR).

### 2.3. Quality Appraisal and Data Extraction

The JBI critical appraisal tools were used to evaluate the quality of the included studies [29]. Studies were assessed in pairs for quality by nine independent reviewers (L.W., A.R., D.L., H.W., I.J., C.S., D.T., F.A. and L.N.). Each item in the quality appraisal was assigned a score of 1, and the overall total score was calculated for each article. Any uncertainty between the reviewers was resolved by discussion to reach an agreement on the final assessment. Data on aims and objectives, methods, study population, nature of the intervention, outcome measures, the phenomena of interest and results were extracted from the included studies.

### 2.4. Data Synthesis

Due to the heterogeneity of study designs used and the outcomes measured, a meta-analysis of the data could not be conducted. Instead, a narrative synthesis of the data is presented according to the Synthesis Without Meta-analysis (SWiM) guidelines [30]. Studies were grouped according to the outcomes reported and study design. Where there was similarity in the outcomes reported and methods of reporting, these findings were compared, and *p*-values, when reported, were included. Where they differed, the outcomes were grouped by type of outcome, e.g., glycaemic control and the direction of the finding reported. Studies of higher methodological quality were reported first in each section. Where findings were reported from lower-quality studies, a sentence to highlight this has been added. The nature of the data reported did not allow for an examination of heterogeneity in reported effects or for an assessment of the certainty of the synthesised findings.

## 3. Results

### 3.1. Study Inclusion

Figure 1 illustrates the flow of studies identified, screened, included, and the reasons for exclusion. A systematic search found a total of 10,449 records. Duplicates (*n* = 4098) were excluded. A total of 6351 records were eligible for the title and abstract screening. Full-text screening of 35 studies yielded 22 eligible studies [20,21,22,23,24,25,31,32,33,34,35,36,37,38,39,40,41,42,43,44,45,46]. After a quality appraisal, seven studies [25,41,42,43,44,45,46] were excluded following a collective agreement between authors that the quality of the studies was too low to merit inclusion. Therefore, 15 studies [20,21,22,23,24,31,32,33,34,35,36,37,38,39,40] were included in the final synthesis.

### 3.2. Characteristics of Included Studies

The review included 14 quantitative studies [20,21,22,23,24,31,32,33,34,35,36,37,38,39] and one qualitative article [40]. Ten studies were quasi-experimental [20,21,22,23,24,31,32,34,35,38], four were RCTs [33,36,37,39] and one was a process evaluation study (qualitative) [40]. Two studies were conducted in New Zealand [39,40], 11 in Iran [20,21,22,23,24,31,32,34,35,36,37] and two in the United States [33,38]. The 15 included studies comprised 684 participants, with sample sizes ranging from 20 [38] to 118 [39]. Twelve studies were conducted in a community setting [20,21,23,24,31,32,33,34,36,38,39,40], two in hospital settings [22,37], and one in a research centre [35]. See Appendix A for an overview of the study characteristics.

### 3.3. Methodological Quality of Included Studies

The assessment of the methodological quality of the studies using the JBI checklist [29] is presented in Table 1, Table 2 and Table 3. Of the ten quasi-experimental studies (maximum quality score 9), four studies were assigned a score of 8 [21,24,31,35], three studies scored 7 [20,22,23], and the remaining studies scored 6 [32] and 5 [34], respectively. There were four RCTs (maximum quality score 13), with two studies scoring 8 [36,37] and two scoring 10 [33,39]. The qualitative study scored 8/10. Two studies [39,40] related to the same sample but reported different outcomes.

### 3.4. Review Findings

Thirteen studies reported ACT as an intervention for people living with T2DM [20,21,23,24,31,33,34,35,36,37,38,39,40]. Only two of the studies included those living with CVD, specifically coronary artery disease [22] and angina pectoris [32]. Eleven studies compared an ACT intervention to a control (standard care or waiting list) group. Three studies [33,37,39] compared an education plus ACT group to an education-only group, and one study [23] compared a healthy lifestyle with ACT group to a mindfulness-based group and a control group.

The interventions ranged from 6.5 h of ACT over one day [39] to 36 h of ACT over 12 sessions [23]. In two studies [20,32], the duration of the intervention sessions was unknown, although most of the ACT interventions were delivered over eight sessions of 90 min duration [21,22,24,34,35,36]. Three studies reported delivering a one day ACT workshop providing 8 [38] or 6.5 [39,40] hours, or an unspecified time period over one day [37] of ACT intervention. All interventions were delivered face-to-face in a group setting [20,21,22,23,24,31,32,33,34,35,36,37,38,39]. Only four studies reported which had delivered ACT interventions. In two studies [39,40], the intervention was delivered by a mental health nurse with expertise in ACT, while Gregg et al. [33] reported that the intervention was delivered by the author of a diabetes and ACT manual. Maghsoudi et al. [36] report that a clinical psychologist and nurse delivered the intervention, while a psychology doctorate student delivered the intervention [38]. Only two studies [33,39] reported expertise or training in ACT before intervention delivery. The primary and secondary outcomes assessed varied and included: glycaemic control [23,33,37,39], general self-management of T2DM [23,33,37,38], quality of life [20,21], stress [20,35], coping strategies [20], acceptance [37], depression [31], emotional distress [36], emotional control [24], knowledge related to diabetes and self-management [40], mental health [34], resilience [35] and self-efficacy [35].

Of the four studies evaluating the impact of the ACT intervention on glycaemic control [23,33,37,39], three observed a significant improvement related to an increase in self-care activities and reduction in glycated haemoglobin [23,33,39]. Four studies evaluated the effect of ACT on self-management behaviour such as weight reduction, dietary changes, exercise plans and glucose monitoring [23,33,37,38], and all reported a significant improvement in diabetes self-care and an increase in the number of people with glycated haemoglobin in the target range [23,33,37,38]. No significant difference in quality of life between the intervention and control groups was reported [20,21], but ACT significantly reduced stress levels in the two studies reporting this outcome [20,35]. A significant improvement in all other outcomes at follow-up were reported, except for those reported by Welch [38] (see Appendix A).

Thirteen studies reported attrition or completion rates, which ranged from 72% completion [39] to 100% [20,21,22,23,24,31,34,35,36]. The qualitative study [40] reported that most participants (66%) described an increase in knowledge around diabetes self-management and an increased sense of personal responsibility. Participants also described changes in self-management activities and reflected on the challenges in instigating and maintaining change to improve diabetes management. The delivery of ACT in a face-to-face group setting was described as acceptable by most participants [40].

## 4. Discussion

This is the first scoping review to explore the research evidence for ACT as an intervention for people living with CVD and/or T2DM. ACT was consistently associated with improved outcomes for people living with CVD or T2DM. There is emerging evidence that ACT may be effective in improving glycaemic control, self-management and stress reduction, with some supportive higher-quality studies. This approach offers promise to support patients and personalise their care regarding their psychological needs, and in doing so, may improve self-management behaviours and clinical outcomes. However, further high-quality research is required on this topic through studies that are adequately powered and well designed, for example, RCTs and process evaluations.

Strengths of this review include the thorough search strategy and the number of full-texts assessed, which included study language translation to ensure inclusivity of all published evidence. The review followed accepted guidelines for conducting a scoping review, and we included a robust process of quality appraisal to acknowledge and raise awareness of the quality of the papers included in the review.

The paucity of studies using RCT designs, small sample sizes (resulting in lack of power), and the generally low quality of studies meant it was unclear whether the findings were due to the intervention, non-specific therapy factors, placebo effects, or regression to the mean. This must be taken into consideration when interpreting the results of the review. While findings to date are encouraging, further high-quality research is needed. Researchers and article authors would benefit from adhering to the reporting standards for intervention studies [48].

In addition to the quality of the studies reported, much variability existed in the sample size, the mode of delivery, the length of the intervention, outcomes measured and reported. The low intensity of the intervention delivery is noteworthy given that all the interventions were delivered as a group, and in this context, may align well into standard diabetes and CVD care approaches. A previous review [14] on ACT across any long-term condition noted the brief administration of ACT sessions as a limitation (highlighting that 58% of interventions included five or fewer sessions). In this review, most of the studies aligned themselves to a clear ACT protocol; however, detail on the implementation of the intervention, explicitly who delivered the ACT intervention and if they were trained to deliver ACT, was poorly reported. None of the studies examined ACT in people living with the comorbidity of heart disease and diabetes. Only one study assessed the acceptability [40] of the intervention. The evidence regarding ACT and outcomes are ambivalent. Most studies were pre-test post-test in design making the assessment of causality challenging. Studies that included a longitudinal or cohort design reported inconclusive results. Furthermore, most of the studies were conducted in Iran, although evidence was included from studies conducted in the United States of America, Australia, and New Zealand. However, the results may not be directly applicable to other cultures and healthcare systems.

## 5. Conclusions

The review reported improved outcomes for people living with CVD or T2DM and that ACT was an acceptable and valid intervention. ACT provides an alternative approach that warrants further assessment in relation to effectiveness. High quality research is needed to further assess the effectiveness of ACT to improve patient reported and clinical outcomes.

## Figures and Tables

**Figure 1 ijerph-18-08126-f001:**
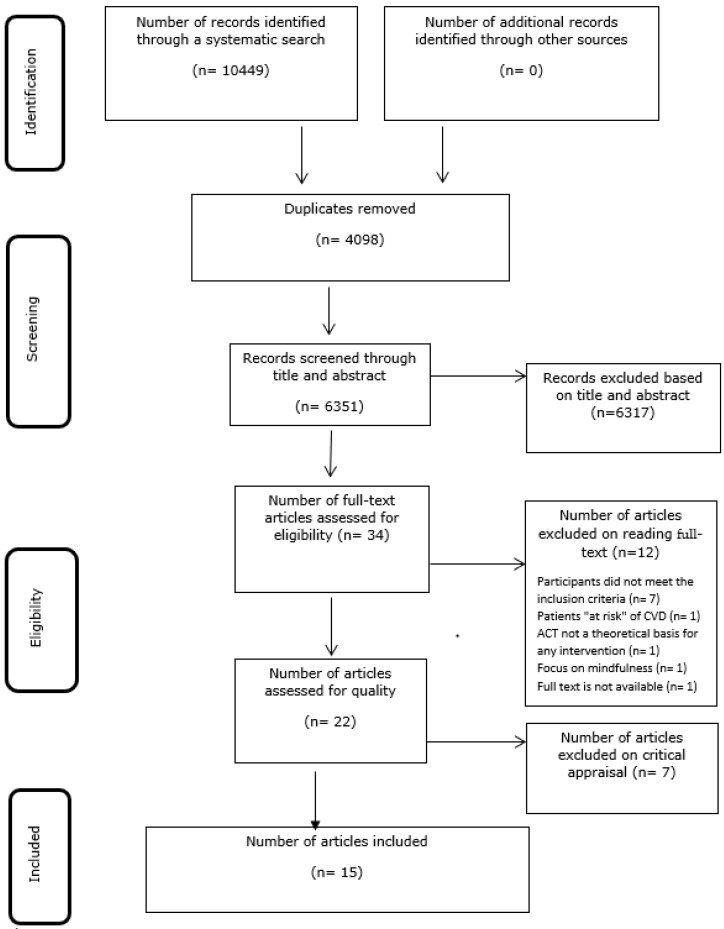
Study Selection and PRISMA flow diagram [47].

**Table 1 ijerph-18-08126-t001:** JBI Critical Appraisal Checklist for Quasi-experimental studies.

Citation	Q1	Q2	Q3	Q4	Q5	Q6	Q7	Q8	Q9	Score /9
[20]	Y	U	U	Y	Y	Y	Y	Y	Y	7/9
[21]	Y	Y	N	Y	Y	Y	Y	Y	Y	8/9
[31]	Y	Y	N	Y	Y	Y	Y	Y	Y	8/9
[22]	Y	Y	U	Y	Y	U	Y	Y	Y	7/9
[32]	Y	U	U	Y	Y	U	Y	Y	Y	6/9
[23]	Y	Y	Y	Y	Y	U	Y	Y	U	7/9
[34]	Y	Y	U	U	Y	U	U	Y	Y	5/9
[35]	Y	Y	N	Y	Y	Y	Y	Y	Y	8/9
[24]	Y	Y	N	Y	Y	Y	Y	Y	Y	8/9
[38]	Y	U	U	N	Y	Y	NA	Y	Y	5/9

Note: Y, yes; N, no; U, unclear; N.A.; not applicable. Questions: 1. Is it clear in the study what is the ‘cause’ and what is the ‘effect’ (i.e., there is no confusion about which variable comes first)? 2. Were the participants, included in any comparisons, similar? 3. Were the participants, included in any comparisons, receiving similar treatment/care, other than the exposure or intervention of interest? 4. Was there a control group? 5. Were there multiple measurements of the outcome both pre- and post- intervention/exposure? 6. Was follow up complete and, if not, were differences between groups in terms of their follow up adequately described and analysed? 7. Were the outcomes of participants included in any comparisons measured in the same way? 8. Were outcomes measured in a reliable way? 9. Was appropriate statistical analysis used?

**Table 2 ijerph-18-08126-t002:** JBI Critical Appraisal Checklist for Qualitative studies.

Citation	Q1	Q2	Q3	Q4	Q5	Q6	Q7	Q8	Q9	Q10	Score /10
[40]	Y	Y	Y	Y	Y	N	U	Y	Y	Y	8/10

Note Y, yes; N, no; U, unclear. Questions: 1. Is there congruity between the stated philosophical perspective and the research methodology? 2. Is there congruity between the research methodology and the research question or objectives? 3. Is there congruity between the research methodology and the methods used to collect data? 4. Is there congruity between the research methodology and the representation and analysis of the data? 5. Is there congruity between the research methodology and the interpretation of the results? 6. Is there a statement locating the researcher culturally or theoretically? 7. Is the influence of the researcher on the research, and vice-versa, addressed? 8. Are participants, and their voices, adequately represented? 9. Is the research ethical, according to current criteria, or for recent studies, and is there evidence of ethical approval by an appropriate body? 10. Do the conclusions drawn in the research report flow from the analysis or interpretation of the data?

**Table 3 ijerph-18-08126-t003:** JBI Critical Appraisal Checklist for randomised controlled trial studies.

Citation	Q1	Q2	Q3	Q4	Q5	Q6	Q7	Q8	Q9	Q10	Q11	Q12	Q13	Score /13
[33]	Y	Y	Y	U	U	U	Y	Y	Y	Y	Y	Y	Y	10/13
[36]	Y	Y	Y	N	N	N	N	Y	Y	Y	Y	Y	Y	8/13
[37]	Y	Y	Y	N	N	N	N	Y	Y	Y	Y	Y	Y	8/13
[39]	Y	Y	Y	N	N	U	Y	Y	Y	Y	Y	Y	Y	10/13

Note Y, yes; N, no; U, unclear; N.A.; not applicable. 1.Was true randomization used for assignment of participants to treatment groups? 2. Was allocation to treatment groups concealed? 3. Were treatment groups similar at the baseline? 4. Were participants blind to treatment assignment? 5. Were outcomes assessors blind to treatment assignment? 6. Were outcomes assessors blind to treatment assignment? 7. Were treatment groups treated identically other than the intervention of interest? 8. Was follow up complete and, if not, were differences between groups in terms of their follow up adequately described and analyzed? 9. Were participants analysed in the groups to which they were randomized? 10. Were outcomes measured in the same way for treatment groups? 11. Were outcomes measured in a reliable way? 12.Was appropriate statistical analysis used?

## Data Availability

All data extracted and synthesised in this review were taken directly from the published articles.

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
