# Peer review of "The Role of Acceptance and Commitment Therapy in Cardiovascular and Diabetes Healthcare: A Scoping Review"

_ijerph, 2021, doi:10.3390/ijerph18158126_

Round 1

Reviewer 1 Report

Include research results that point to possible causes of heterogeneity between study results. And other studies applying RCT designs, with larger sample sizes.

Author Response

Letter in response to reviewer’s comments

Manuscript ID ijerph-1255563

Title: The role of acceptance and commitment therapy in cardiovascular and diabetes healthcare: a scoping review

Dear Kayla,

We would like to thank the Associate Managing Editor and the reviewers for their careful and thorough consideration of this manuscript and for the thoughtful comments and constructive suggestions for revision. Below we address each of their suggestions in turn.  

We believe that our revised manuscript is much improved and will be of great interest to readers of the International Journal of Environmental Research and Public Health

Reviewer 1

Include research results that point to possible causes of heterogeneity between study results. And other studies applying RCT designs, with larger sample sizes.

Response: Thank you. The intent of this scoping review was to include a range of literature available on the use of ACT rather than focus on RCTs alone. Where no review has drawn together the available evidence to date we felt this was important. The sources of heterogeneity between the studies have been reported in the results section and the following sentence has been added to help highlight these areas for clearly for the reader.

“In this section we report on study design, characteristics of the intervention by study and the primary and secondary outcomes. An overview of the study design and outcomes by study are presented in Appendix 1”

Reviewer 2 Report

The main objective of the work is to carry out a systematic evaluation on the efficacy of acceptance and commitment therapy.

The objective is pertinent since implementing effective psychological therapies that help to manage and improve the impact of physical illnesses is one of the challenges of public health policies.

The authors use a specific methodology (Joanna Briggs Institute guidelines) to carry out the review.

The methodology is adequate to meet the objectives and is correctly explained.

Conclusions are consistently derived from the results.

Minor comments.

You need to check the references. Some citations contain errors. For example: reference 38 line 419 or reference 39 line 422.

Author Response

Letter in response to reviewer’s comments

Manuscript ID ijerph-1255563

Title: The role of acceptance and commitment therapy in cardiovascular and diabetes healthcare: a scoping review

Dear Kayla,

We would like to thank the Associate Managing Editor and the reviewers for their careful and thorough consideration of this manuscript and for the thoughtful comments and constructive suggestions for revision. Below we address each of their suggestions in turn.  

We believe that our revised manuscript is much improved and will be of great interest to readers of the International Journal of Environmental Research and Public Health

The main objective of the work is to carry out a systematic evaluation on the efficacy of acceptance and commitment therapy. The objective is pertinent since implementing effective psychological therapies that help to manage and improve the impact of physical illnesses is one of the challenges of public health policies. The authors use a specific methodology (Joanna Briggs Institute guidelines) to carry out the review. The methodology is adequate to meet the objectives and is correctly explained. Conclusions are consistently derived from the results.

Response: Thank you very much.

You need to check the references. Some citations contain errors. For example: reference 38 line 419 or reference 39 line 422.

Response: Thank you very much. As suggested by the reviewer, changes have been made and added  in the revised manuscript.

Reviewer 3 Report

Authors investigated to the role of ACT in CVD and DM patients. It is difficult to evaluate the ACT evidence from this review.

  1. Introduction:
    1. The patients’ backgrounds between CVD and T2DM are quite different. Authors should investigate separately. In addition, the background of CVD is unclear.
    2. The definition and profits of ACT are insufficient to understand.
  2. Methods
    1. Study design is inadequate. Authors included low quality studies and many biases.
  3. Results
    1. Since authors include a lot of outcomes at the same time, it is difficult to understand. Please mention individually.
  4. Conclusion does not reflect on results.

Author Response

Letter in response to reviewer’s comments

Manuscript ID ijerph-1255563

Title: The role of acceptance and commitment therapy in cardiovascular and diabetes healthcare: a scoping review

Dear Kayla,

We would like to thank the Associate Managing Editor and the reviewers for their careful and thorough consideration of this manuscript and for the thoughtful comments and constructive suggestions for revision. Below we address each of their suggestions in turn.  

We believe that our revised manuscript is much improved and will be of great interest to readers of the International Journal of Environmental Research and Public Health

Reviewer 3

Introduction: The patients’ backgrounds between CVD and T2DM are quite different. Authors should investigate separately. In addition, the background of CVD is unclear.

Response: Thank you for your feedback. The following sentences have been added to the introduction to highlight the shared risk factors and the prevalence of co-morbidity. The two conditions share many similarities. In this paper we do not provide an overview of the pathophysiology of either disease but instead an overview of the prevalence and the impact at a societal level.

“Diabetes is a risk factor for cardiovascular disease and there is a high prevalence of co-morbidity”.

The definition and profits of ACT are insufficient to understand.

Response: Thank you for your feedback. The section introducing ACT now reads:

Acceptance and Commitment Therapy (ACT) is an adapted form of Cognitive Behavioural Therapy. Unlike Cognitive Behavioural Therapy (CBT) and traditional behavioural-based approaches, ACT focuses on how thinking affects behaviour, and promotes psychological flexibility. In comparison to CBT, ACT does not attempt to change beliefs (e.g. replace negative thoughts with adaptive thoughts), and it does not seek to remove the psychological distress (though this may be a positive outcome). ACT promotes the connection to the present moment and engagement in values-based action to help people create a sense of meaning and increase their psychological flexibility. (Hayes et al., 2012).

Methods: Study design is inadequate. Authors included low quality studies and many biases.

Response: Thank you for your feedback. We excluded seven studies based on methodological quality (please see section 3.1 lines 150-152). The studies included in the review did vary in quality and we have taken care to highlight this throughout the paper as well as reporting results by methodological quality.

Results: Since authors include a lot of outcomes at the same time, it is difficult to understand. Please mention individually.

Response: Thank you: We have added the following sentence to provide an overview of the key areas covered in the results section and also refer the reader to Appendix 1 which sets out a summary of the characteristics including outcomes reported.

“In this section we report on study design, characteristics of the intervention by study and the primary and secondary outcomes. An overview of the study design and outcomes by study are presented in Appendix 1”

Conclusion does not reflect on results.

Response: Thank you for your feedback. The following sentence has been added to the revised manuscript (please see pages 12) as follows:

“The review reported improved outcomes for people living with CVD or T2DM and that ACT was an acceptable and valid intervention. ACT provides an alternative approach that warrants further assessment in relation to effectiveness. High quality research is needed to further assess the effectiveness of ACT to improve patient reported and clinical outcomes”.

Round 2

Reviewer 3 Report

Authors answered my questions and corrected well.